# Confidence Difference Reflects Various Supervised Signals in Confidence-Difference Classification

**Yuanchao Dai** [1 2]  **Ximing Li** [1 2]  **Changchun Li** [1 2]

## Abstract

Training a precise binary classifier with limited supervision in weakly supervised learning scenarios holds considerable research significance in practical settings. Leveraging pairwise unlabeled data with confidence differences has been demonstrated to outperform learning from pointwise unlabeled data. We theoretically analyze the various supervisory signals reflected by confidence differences in confidence difference (ConfDiff) classification and identify challenges arising from noisy signals when confidence differences are small. To address this, we partition the dataset into two subsets with distinct supervisory signals and propose a consistency regularization-based risk estimator to encourage similar outputs for similar instances, mitigating the impact of noisy supervision. We further derive and analyze its estimation error bounds theoretically. Extensive experiments on benchmark and UCI datasets demonstrate the effectiveness of our method. Additionally, to effectively capture the influence of real-world noise on the confidence difference, we artificially perturb the confidence difference distribution and demonstrate the robustness of our method under noisy conditions through comprehensive experiments.

## 1. Introduction

Weakly supervised learning is an essential research field in machine learning, focusing on training accurate predictive models under conditions of low supervision or imprecise labeling. Due to the difficulty of obtaining precise supervision in real-world scenarios, weakly supervised learning holds

significant research value for effectively leveraging limited available supervision information. Consequently, the field of weakly supervised learning has increasingly attracted attention from experts and scholars in recent years, leading to the emergence of many typical weakly supervised learning methods, such as multi-instance learning (Zhou et al., 2009; Zhang & Zhou, 2009; Wu et al., 2018; Shi et al., 2020), positive and unlabeled (PU) learning (Kiryo et al., 2017; Hammoudeh & Lowd, 2020; Zhao et al., 2022; Luo et al., 2021; Wang et al., 2023), and others.

A prevalent idea in weakly supervised classification involves maximizing the utilization of pointwise weakly supervised information (Feng et al., 2021), thereby prompting the development of various techniques based on soft labels (Nguyen et al., 2014; Xue & Hauskrecht, 2016), mixup (Zhang et al., 2017; Verma et al., 2019; Yun et al., 2019; Kim et al., 2020; Hendrycks et al., 2019; Li et al., 2021), and others. Nevertheless, it is undeniable that annotating pointwise information in real-world classification problems is a complex and laborious task, further compounded by the personal biases of annotators which frequently exacerbate the probability of inaccuracies. In such scenarios, pairwise comparison information between data points may be more readily obtainable in real-world settings than pointwise information, and it often exhibits greater resistance to biases compared to pointwise semi-supervised information (Bao et al., 2018). For instance, in medical diagnosis, accurately determining whether a patient has a disease solely based on their presented symptoms is challenging. However, comparing the symptoms of this patient with those of others provides more accessible information and reduces the probability of misdiagnosis. Extensive research has been conducted on pairwise analysis in numerous binary classification problems, leading to the development of risk minimization functions capable of inducing binary classifiers across various combinations of pairwise similarities, dissimilarities, and unlabeled data (Bao et al., 2018; Shimada et al., 2021; Lu et al., 2018; 2020; Wang et al., 2024).

In recent work, pairwise comparison (Pcomp) classification has shown that in tackling difficult point labeling tasks, people can more easily gather comparative information between two instances, constituting a form of weakly supervised information (Feng et al., 2021). However, in real-world appli-

[1] College of Computer Science and Technology, Jilin University, China [2] Key Laboratory of Symbolic Computation and Knowledge Engineering of Ministry of Education, Jilin University, China. Correspondence to: Ximing Li <liximing86@gmail.com>.

cation scenarios, individuals may not only distinguish which of two instances is more likely to be classified as positive over the other but also gauge the extent of the disparity in their confidence levels regarding positivity. In light of this framework, Wang *et al.* introduce a new pairwise weakly supervised classification problem called confidence-difference (ConfDiff) classification, and propose the corresponding ConfDiff method (Wang et al., 2024). To establish confidence difference, the ConfDiff method first utilizes binary-labeled data to train a probability classifier. Subsequently, pairwise instances are fed into the classifier to generate posterior probabilities, from which confidence differences are computed based on the differences between these posterior probabilities. However, through analysis of the various supervised signals in the ConfDiff method, we identify that the method encourages models to predict opposite classes for pairwise instances, as supported by both experimental and theoretical perspectives. This prediction direction is valid when the confidence difference is large. However, when the confidence difference is small, the instances may belong to either the same or different classes, and such a predictive tendency may lead the model to incorrectly classify instances from the same class as belonging to different classes, thereby introducing less reliable supervised signals.

To handle this problem, in this paper, we concentrate on mitigating the impact of these less reliable supervised signals when confidence differences are small. Specifically, we analyze the different supervised signals induced by varying confidence differences in the ConfDiff method (Wang et al., 2024). We find that pairwise instances with small confidence differences tend to introduce less reliable supervised signals, while those with larger confidence differences provide more reliable supervision. Based on this observation, we propose a ConfDiff classification method that incorporates consistency regularization. By partitioning the dataset based on the reliability of supervised signals, we introduce a consistency regularization term for the subset with less reliable supervised signals, encouraging the model to produce similar outputs for pairwise instances with small confidence differences. Meanwhile, for the subset with more reliable supervised signals, we preserve the benefit of these reliable supervised signals. Experimental results demonstrate that our method outperforms existing baselines in most cases and exhibits strong robustness even under artificial noise interference.

In summary, this paper's key contributions can be outlined as follows:

- We introduce a method for ConfDiff classification which aims to enhance the accuracy of weakly supervised classification by constructing risk estimator through **C**onsistency **R**isk and **C**onsistency **R**egularization (CRCR).

- We theoretically analyze various supervised signals reflected by different confidence differences in ConfDiff classification. Additionally, we theoretically estimate the error bounds of our proposed method.

- We validate the effectiveness of our method through experiments by comparing it with existing baselines on datasets of varying scales. In addition, the robustness of our method is further validated under the influence of artificially added noise.

## 2. Preliminaries

In this section, we briefly review the problem definitions of binary classification, binary classification with soft labels, and ConfDiff classification.

**Formulation of binary classification**  Binary classification is a typical task in the field of supervised learning, where the goal is to induce a classifier that partitions the data space into two categories. Formally, let $\mathcal{X} = \mathbb{R}^d$ and $\mathcal{Y} = \{-1, +1\}$ be the $d$-dimensional feature space and label space, respectively. The dataset $\mathcal{D}_{\mathrm{BC}} = \mathcal{D}_{\mathrm{BC}}^p \cup \mathcal{D}_{\mathrm{BC}}^n$ for binary classification consists of a positive dataset $\mathcal{D}_{\mathrm{BC}}^p$ and a negative dataset $\mathcal{D}_{\mathrm{BC}}^n$:

$$\mathcal{D}_{\mathrm{BC}}^p = \{(\mathbf{x}_i^p \in \mathcal{X}, y_i^p = +1)\}_{i=1}^{n_p}, \ \mathbf{x}_i^p \overset{i.i.d.}{\sim} p(\mathbf{x}|y = +1),$$

$$\mathcal{D}_{\mathrm{BC}}^n = \{(\mathbf{x}_i^n \in \mathcal{X}, y_i^n = -1)\}_{i=1}^{n_n}, \ \mathbf{x}_i^n \overset{i.i.d.}{\sim} p(\mathbf{x}|y = -1),$$

where $n_p$ and $n_n$ denote the number of positive and negative instances, respectively. Let $\pi$ denote the positive class prior $p(y = +1)$ and $\ell : \mathbb{R} \times \mathcal{Y} \to \mathbb{R}_+$ be a binary loss function. Then binary classification induces a classifier $g : \mathcal{X} \to \mathbb{R}$ from $\mathcal{D}_{\mathrm{BC}}$ by minimizing the following classification risk:

$$\begin{aligned} R(g) =& \pi \mathbb{E}_{p(\mathbf{x}|y=+1)}[\ell\big(g(\mathbf{x}), +1\big)] \\ &+ (1-\pi)\mathbb{E}_{p(\mathbf{x}|y=-1)}[\ell\big(g(\mathbf{x}), -1\big)]. \end{aligned} \quad (1)$$

**Formulation of binary classification with soft labels**  In binary classification, soft labels typically represent the confidence of each sample belonging to the positive class. Moreover, several studies have shown that using soft labels rather than hard labels can more accurately reflect the data distribution (Szegedy et al., 2016), thus enhancing the accuracy of training binary classifiers. Formally, let $q_i$ denote the positive confidence of $\mathbf{x}_i$, the dataset $\mathcal{D}_{\mathrm{BC\text{-}soft}}$ for binary classification can be defined as follows:

$$\mathcal{D}_{\mathrm{BC\text{-}soft}} = \{(\mathbf{x}_i, q_i)\}_{i=1}^n, \ \mathbf{x}_i \overset{i.i.d.}{\sim} p(\mathbf{x}),$$
$$q_i = p(y_i = +1|\mathbf{x}_i),$$

where $p(\mathbf{x}) = \pi p(\mathbf{x}|y = +1) + (1 - \pi)p(\mathbf{x}|y = -1)$. Subsequently, the risk minimization objective function for

binary classification with soft labels can be reformulated into the following form:

$$R_{\text{BC-soft}}(g) = \mathbb{E}_{p(\mathbf{x})}[q\ell(g(\mathbf{x}), +1) + (1 - q)\ell(g(\mathbf{x}), -1)]. \tag{2}$$

**Formulation of confidence-difference (ConfDiff) classification** Given that pairwise supervision is typically more accessible than pointwise supervision and it's feasible not only to determine which sample in an unlabeled data pair is more likely positive but also quantify the confidence difference between them in practical scenarios, ConfDiff classification precisely serves as a weakly supervised classification tailored to address this scenario. It specifically deals with weakly supervised classification problems where training data comprises only pairwise unlabeled data and the confidence difference associated with each pair. Formally, let $c_i = c(\mathbf{x}_i, \mathbf{x}_i') = p(y_i' = +1|\mathbf{x}_i') - p(y_i = +1|\mathbf{x}_i)$ be the confidence difference between pairwise unlabeled data $(\mathbf{x}_i, \mathbf{x}_i')$ drawn from an independent and identically distributed probability density $p(\mathbf{x}, \mathbf{x}') = p(\mathbf{x})p(\mathbf{x}')$. Let $\mathcal{D}_{\text{CD}}$ denote a pairwise dataset drawn from the pairwise unlabeled data and the confidence differences between them:

$$\mathcal{D}_{\text{CD}} = \{((\mathbf{x}_i, \mathbf{x}_i'), c_i)\}_{i=1}^n, \ \mathbf{x}_i \overset{i.i.d.}{\sim} p(\mathbf{x}), \ \mathbf{x}_i' \overset{i.i.d.}{\sim} p(\mathbf{x}).$$

Recent studies deal with the ConfDiff classification problem in such challenging scenarios (Wang et al., 2024). They deduce an unbiased risk estimator for confidence-difference classification from Eq.1 and train a binary classifier solely utilizing unlabeled data and confidence differences by minimizing it. The classification risk can be expressed as:

$$R_{\text{CD}}(g) = \frac{1}{2}\mathbb{E}_{p(\mathbf{x}, \mathbf{x}')}[\mathcal{L}(\mathbf{x}, \mathbf{x}') + \mathcal{L}(\mathbf{x}', \mathbf{x})], \tag{3}$$

where $\mathcal{L}(\mathbf{x}, \mathbf{x}') = (\pi - c(\mathbf{x}, \mathbf{x}'))\ell(g(\mathbf{x}), +1) + (1 - \pi - c(\mathbf{x}, \mathbf{x}'))\ell(g(\mathbf{x}'), -1)$. Then Eq.3 can be refined as follows:

$$\begin{aligned}R_{\text{CD}}(g) = \frac{1}{2}\mathbb{E}_{p(\mathbf{x}, \mathbf{x}')}[&(\pi - c(\mathbf{x}, \mathbf{x}'))\ell(g(\mathbf{x}), +1) \\ &+ (1 - \pi - c(\mathbf{x}, \mathbf{x}'))\ell(g(\mathbf{x}'), -1) \\ &+ (\pi + c(\mathbf{x}, \mathbf{x}'))\ell(g(\mathbf{x}'), +1) \\ &+ (1 - \pi + c(\mathbf{x}, \mathbf{x}'))\ell(g(\mathbf{x}), -1)]. \end{aligned} \tag{4}$$

## 3. The Proposed Method

In this section, we introduce the proposed ConfDiff method.

### 3.1. Analysis of the ConfDiff Method

In the ConfDiff method, pairwise instances with small confidence differences $|c(\mathbf{x}, \mathbf{x}')|$ are prone to introducing less reliable supervised signals, while those with larger confidence

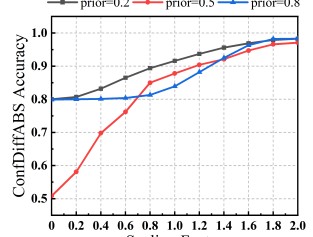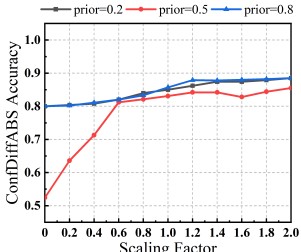

*Figure 1.* The Accuracy for the binary classifier about different proportion of pairwise data with $|c(\mathbf{x}, \mathbf{x}')| > 0.5$ on two benchmark datasets MNIST (left) and CIFAR-10 (right). (The value of the x-axis values $* \min(\pi, 1 - \pi)$ denotes the proportion of pairwise instances with $|c(\mathbf{x}, \mathbf{x}')| > 0.5$.)

differences $|c(\mathbf{x}, \mathbf{x}')|$ are considered to provide stronger and more reliable supervised signals.

To explain this, we consider the general form of many commonly used losses for the prediction function $g(\mathbf{x})$ and target $y$ (Zhang et al., 2021):

$$\begin{aligned}\mathcal{L} = \{\ell(g(\mathbf{x}), y)|\ell(g(\mathbf{x}), y) = h(g(\mathbf{x})) - yg(\mathbf{x}) \\ \text{for some function } h\}. \end{aligned} \tag{5}$$

Substituting the form of the loss function from Eq.5 into Eq.4, then the classification risk of ConfDiff method can be rewritten as follows and the proof details are presented in the Appendix B:

$$\begin{aligned}R_{\text{CD}}(g) = &\frac{1}{2}\mathbb{E}_{p(\mathbf{x}, \mathbf{x}')}\Big[\big(\frac{1}{2} - c(\mathbf{x}, \mathbf{x}')\big)\ell(g(\mathbf{x}), +1) \\ &\qquad + \big(\frac{1}{2} + c(\mathbf{x}, \mathbf{x}')\big)\ell(g(\mathbf{x}'), +1)\Big] \\ &+ \frac{1}{2}\mathbb{E}_{p(\mathbf{x}, \mathbf{x}')}\Big[\big(\frac{1}{2} + c(\mathbf{x}, \mathbf{x}')\big)\ell(g(\mathbf{x}), -1) \\ &\qquad + \big(\frac{1}{2} - c(\mathbf{x}, \mathbf{x}')\big)\ell(g(\mathbf{x}'), -1)\Big] \\ &+ \frac{1}{2}\mathbb{E}_{p(\mathbf{x}, \mathbf{x}')}\Big[(1 - 2\pi)(g(\mathbf{x}) + g(\mathbf{x}'))\Big], \end{aligned} \tag{6}$$

where the first and second terms denote the pairwise instance $(\mathbf{x}, \mathbf{x}')$ contrastive losses for positive and negative class predictions, respectively; while the third term serves as a regularization. We first analyze the critical components of the first term, where the weights $\frac{1}{2} - c(\mathbf{x}, \mathbf{x}')$ and $\frac{1}{2} + c(\mathbf{x}, \mathbf{x}')$ determine the contributions of $\mathbf{x}$ and $\mathbf{x}'$ to the positive class prediction loss, respectively. These weights demonstrate a natural balance as they sum to 1, with $\frac{1}{2}$ acting as a critical threshold that determines prediction directionality. Since the weights $\frac{1}{2} - c(\mathbf{x}, \mathbf{x}')$ and $\frac{1}{2} + c(\mathbf{x}, \mathbf{x}')$ necessarily fall on opposite sides of this threshold, they create a contrastive learning dynamic: when one instance is pushed toward stronger positive class prediction, the other

is simultaneously pushed toward weaker positive class prediction (effectively toward the negative class). In other words, the first loss term ensures that $\mathbf{x}$ and $\mathbf{x}'$ adjust their predictions in opposite directions, thereby emphasizing the predictive divergence of pairwise instances in the positive class predictions. Similarly, the second loss term forces the instances to diverge in their predictions for the negative class.

Referring to the definition of $c(\mathbf{x}, \mathbf{x}')$, the magnitude of confidence difference $|c(\mathbf{x}, \mathbf{x}')|$ naturally reflects the reliability of the supervised signal between pairwise instances. When $|c(\mathbf{x}, \mathbf{x}')|$ is large, there is a strong indication that $\mathbf{x}$ and $\mathbf{x}'$ likely belong to different classes, as their posterior probability difference exceeds the classification threshold. Conversely, when $|c(\mathbf{x}, \mathbf{x}')|$ is small, the supervised signal becomes less reliable; $\mathbf{x}$ and $\mathbf{x}'$ may belong to either the same class or different classes, as the posterior difference is insufficient to provide a precise classification signal. Therefore, we consider that pairwise instances with larger $|c(\mathbf{x}, \mathbf{x}')|$ tend to provide more reliable supervised signals, while those with smaller $|c(\mathbf{x}, \mathbf{x}')|$ may contribute less reliable supervised signals in the existing ConfDiff method.

To further validate this perspective, we conduct experiments on MNIST and CIFAR-10 datasets by varying the proportion of the pairwise instances with $|c(\mathbf{x}, \mathbf{x}')| > 0.5$, as 0.5 corresponds to the natural classification decision boundary where the posterior probability difference equals the binary classification threshold. The empirical results (shown in Figure 1) illustrate the accuracy under different proportions of the pairwise instances with $|c(\mathbf{x}, \mathbf{x}')| > 0.5$. We observe a positive correlation between classification accuracy and the proportion value. Notably, when the proportion is 0, the accuracy is approximately 0.5, indicating that the classifier performs nearly at random. These findings demonstrate that the pairwise instances with $|c(\mathbf{x}, \mathbf{x}')| > 0.5$ provide stronger and more reliable supervised signals and dominate the contribution to $R_{\text{CD}}$.

### 3.2. CRCR Method

Based on the discussion in Section 3.1, it has been demonstrated that pairwise instances with larger $|c(\mathbf{x}, \mathbf{x}')|$ tend to provide more reliable supervised signals, while those with smaller $|c(\mathbf{x}, \mathbf{x}')|$ provide less reliable supervised signals. To address this issue, we propose setting a threshold $\theta$ to partition the dataset into two subsets: one containing pairwise instances with $|c(\mathbf{x}, \mathbf{x}')| > \theta$ (more reliable supervised signals, denoted as $D^S$) and the other with $|c(\mathbf{x}, \mathbf{x}')| \leq \theta$ (less reliable supervised signals, denoted as $D^C$). For $D^C$, we aim to provide additional guidance to direct the predictions of pairwise instances toward more appropriate outcomes. Specifically, for pairwise instances with smaller $|c(\mathbf{x}, \mathbf{x}')|$, we encourage the model to produce more similar outputs

for these pairs, as they likely belong to the same class. To achieve this, we introduce a consistency regularization term that promotes alignment between the confidence differences and the model's outputs. Meanwhile, for $D^S$, we retain the original strategy to preserve the effectiveness of the predictions driven by this strong guidance.

Our objective is to induce a classifier $g\colon \mathbb{R}^d \to \mathcal{Y}$ from $\mathcal{D}$ by minimizing the expected risk with respect to the data distribution:

$$
\begin{aligned}
&R_{\text{CRCR}}(g) \\
&= \frac{1}{2}\mathbb{E}_{p_{\mathcal{D}^S}(\mathbf{x}, \mathbf{x}')}\Big[\big(\pi - c(\mathbf{x}, \mathbf{x}')\big)\ell\big(g(\mathbf{x}), +1\big) \\
&\qquad\qquad + \big(1 - \pi - c(\mathbf{x}, \mathbf{x}')\big)\ell\big(g(\mathbf{x}'), -1\big) \\
&\qquad\qquad + \big(\pi + c(\mathbf{x}, \mathbf{x}')\big)\ell\big(g(\mathbf{x}'), +1\big) \\
&\qquad\qquad + \big(1 - \pi + c(\mathbf{x}, \mathbf{x}')\big)\ell\big(g(\mathbf{x}), -1\big)\Big] \\
&\quad + \alpha\mathbb{E}_{p_{\mathcal{D}^C}(\mathbf{x}, \mathbf{x}')}\Big[\big(\frac{1}{\log\left(|c(\mathbf{x}, \mathbf{x}')| + \varepsilon\right)}\big) \cdot \|g(\mathbf{x}) - g(\mathbf{x}')\|_2\Big],
\end{aligned}
\tag{7}
$$

where $\alpha$ denotes the parameter of the consistency regularization term, and $\varepsilon = 1.1$ is a smoothing parameter introduced to mitigate numerical issues when $|c(\mathbf{x}, \mathbf{x}')|$ approaches or equals zero. Let $\left|\mathcal{D}^S\right| = n_1$ and $\left|\mathcal{D}^C\right| = n_2$. Then the empirical risk estimator can be expressed as follows:

$$
\begin{aligned}
&\hat{R}_{\text{CRCR}}(g) \\
&= \frac{1}{2n_1}\sum_{i=1}^{n_1}\Big(\big(\pi - c_i\big)\ell\big(g(\mathbf{x}_i), +1\big) \\
&\quad + (1 - \pi - c_i)\ell\big(g(\mathbf{x}'_i), -1\big) + (\pi + c_i)\ell\big(g(\mathbf{x}'_i), +1\big) \\
&\quad + (1 - \pi + c_i)\ell\big(g(\mathbf{x}_i), -1\big)\Big) \\
&\quad + \frac{\alpha}{n_2}\sum_{i=1}^{n_2}\left(\frac{1}{\log\left(|c_i| + \varepsilon\right)} \cdot \|g(\mathbf{x}_i) - g(\mathbf{x}'_i)\|_2\right). \tag{8}
\end{aligned}
$$

### 3.3. Analysis of Error Bound

Assuming there exists a constant $C_g$ such that $\sup_{g \in G}\|G\|_\infty \leq C_g$, and another constant $C_\ell$ such that $\sup_{|z| \leq C_g}\ell(z, y) \leq C_\ell$. Additionally, we presume the binary loss function $\ell(z, y)$ to be Lipschitz continuous with respect to both $z$ and $y$, and to have a Lipschitz constant denoted by $L_\ell$. $\mathfrak{R}_{n_1}(\mathcal{G})$ and $\mathfrak{R}_{n_2}(\mathcal{G})$ denote the Rademacher complexity of unlabeled data $\mathcal{G}$ with size $n_1$ and $n_2$, respectively.

**Theorem 3.1.** *Let $g^* = \arg\min_{g \in \mathcal{G}} R(g)$ be the minimizer of the true classification risk in Eq.1 and $\hat{g}_{\text{CRCR}} = \arg\min_{g \in \mathcal{G}}\hat{R}_{\text{CRCR}}(g)$ denotes the minimizer of the risk form in Eq.8. Then for any $\delta > 0$, we believe that the*

*following expression holds with a probability at least $1 - \delta$:*

$$R(\hat{g}_{\text{CRCR}}) - R(g^*) \leq (8 + 4\beta)L_\ell\mathfrak{R}_{n_1}(\mathcal{G})$$

$$+ \frac{6\alpha}{\log(\varepsilon)}\mathfrak{R}_{n_2}(\mathcal{G}) + 2C_\ell\sqrt{\frac{\ln(2/\delta)}{2n_1}}. \quad (9)$$

Due to the space limitation, the proof details are presented in Appendix A. As $n_1, n_2 \to \infty$, the Rademacher complexities $\mathfrak{R}_{n_1}(\mathcal{G})$ and $\mathfrak{R}_{n_2}(\mathcal{G})$ decrease to zero, and the third term involving $1/\sqrt{n_1}$ also diminishes. Furthermore, the convergence rates of $\mathfrak{R}_{n_1}(\mathcal{G})$ and $\mathfrak{R}_{n_2}(\mathcal{G})$ are $O(1/\sqrt{n_1})$ and $O(1/\sqrt{n_2})$, while the third term's rate is dominated by $O(1/\sqrt{n_1})$. Consequently, as $n \to \infty$, $R(\hat{g}_{\text{CRCR}}) \to R(g^*)$, and the overall convergence rate is characterized by $O\left(\max(1/\sqrt{n_1}, 1/\sqrt{n_2})\right)$.

### 3.4. Empirical Risk Correction

It can potentially lead to severe overfitting problems when the empirical risk becomes negative due to the application of an unbiased risk estimator. Fortunately, risk correction functions $f(\cdot)$ can be utilized to mitigate this issue. Examples include the absolute value function or the rectified linear unit (ReLU) function. Consequently, the corrected risk estimator can be expressed as follows:

$$\tilde{R}_{\text{CRCR}}(g) = \frac{1}{2n_1}f\left(\sum_{i=1}^{n_1}(\pi - c_i)\ell\big(g(\mathbf{x}_i), +1\big)\right)$$

$$+ \frac{1}{2n_1}f\left(\sum_{i=1}^{n_1}(1 - \pi - c_i)\ell\big(g(\mathbf{x}'_i), -1\big)\right)$$

$$+ \frac{1}{2n_1}f\left(\sum_{i=1}^{n_1}(\pi + c_i)\ell\big(g(\mathbf{x}'_i), +1\big)\right)$$

$$+ \frac{1}{2n_1}f\left(\sum_{i=1}^{n_1}(1 - \pi + c_i)\ell\big(g(\mathbf{x}_i), -1\big)\right)$$

$$+ \alpha\frac{1}{n_2}f\left(\sum_{i=1}^{n_2}\left(\frac{1}{\log\left(|c_i| + \varepsilon\right)} \cdot \|(g(\mathbf{x}_i) - g(\mathbf{x}'_i)\|_2\right)\right). \quad (10)$$

Additionally, in our experiments, we report results for two variants of our method that utilizes the absolute value risk correction function (CRCR-ABS) and ReLU risk correction function (CRCR-ReLU).

## 4. Experiments

In this section, we empirically evaluate the proposed CRCR method.

### 4.1. Experimental Settings

**Datasets**    To thoroughly evaluate our method, we employ four popular benchmark datasets, including MNIST (Le-Cun et al., 1998), Kuzushiji-MNIST (K-MNIST) (Clanuwat et al., 2018), Fashion-MNIST (F-MNIST)(Xiao et al., 2017) and CIFAR-10 (Krizhevsky, Technical report, University of Toronto, 2009). Additionally, experiments are conducted on two UCI datasets [1], including Optdigits and Pendigits. Since these datasets contain multiple classes, we categorize the class labels into positive and negative classes, effectively transforming them into binary classification datasets. Furthermore, for each dataset, we randomly select $m\% \times n$ instances to artificially add noise, where the noise ratio $m$ is varied over $[0, 50, 75, 100]$. As a result, in our experiments, we generate 24 synthetic datasets in total.

Moreover, we choose different models as backbones based on the varying feature dimensions of each dataset. Specifically, for MNIST, K-MNIST and F-MNIST, we use a 3-layer multilayer perceptron (MLP) with three hidden layers of width 300 equipped with the ReLU (Nair & Hinton, 2010) activation function and batch normalization (Ioffe & Szegedy, 2015). For CIFAR-10, we train a ResNet-34 model (He et al., 2016) as the backbone. For all UCI datasets, we use a linear model for training. The detailed information for each dataset is presented in Table 1.

**Baseline methods**    We employ seven state-of-the-art algorithms for comparison, including four Pcomp methods (*i.e.,*, Pcomp-Teacher, Pcomp-ABS, Pcomp-ReLU and Pcomp-Unbiased) and three ConfDiff methods (*i.e.,*, ConfDiff-ABS, ConfDiff-ReLU and ConfDiff-Unbiased). Details of baselines are described as follows:

- Pointwise Binary Classification with **P**airwise Confidence **Comp**arisons (**Pcomp**) (Feng et al., 2021): A weakly supervised learning method that trains a binary classifier using pairwise comparison data, composed of unlabeled data pairs where one is more likely to be positive, instead of using pointwise data. Pcomp comprises four versions: Pcomp-Teacher, Pcomp-ABS, Pcomp-ReLU, and Pcomp-Unbiased. We use the code provided by its authors [2].

- Binary Classification with **Conf**idence **Diff**erence (**ConfDiff**) (Wang et al., 2024): A weakly supervised learning method that trains a binary classifier using pairwise comparison data, which consists of pairwise unlabeled data where the difference in the probabilities of being positive (confidence difference) is known. ConfDiff comprises three versions: ConfDiff-ABS, ConfDiff-ReLU, and ConfDiff-Unbiased. We utilize the publicly available code online[3].

---

[1]http://archive.ics.uci.edu/
[2]https://lfeng1995.github.io/codedata.html
[3]https://github.com/wwangwitsel/ConfDiff

*Table 1.* Detailed characteristics of datasets.

| Dataset | #Instance | #Trainset | #Testset | #Fea | Pos Class | Neg Class | Backbone |
|---------|-----------|-----------|----------|------|-----------|-----------|----------|
| MNIST | 70,000 | 15,000 | 5,000 | $28 \times 28$ | 0,2,4,6,8 | 1,3,5,7,9 | 3-layer MLP |
| F-MNIST | 70,000 | 15,000 | 5,000 | $28 \times 28$ | 0,2,4,6,8 | 1,3,5,7,9 | 3-layer MLP |
| K-MNIST | 70,000 | 15,000 | 5,000 | $28 \times 28$ | 0,2,4,6,8 | 1,3,5,7,9 | 3-layer MLP |
| CIFAR-10 | 60,000 | 10,000 | 5,000 | $3 \times 32 \times 32$ | 2,3,4,5,6,7 | 0,1,8,9 | ResNet-34 |
| Optdigits | 5,620 | 1,200 | 1,125 | 62 | 0,2,4,6,8 | 1,3,5,7,9 | Linear |
| Pendigits | 10,992 | 2,500 | 2,199 | 16 | 0,2,4,6,8 | 1,3,5,7,9 | Linear |

**Implementation details** For each comparison method under every experimental configuration, we execute the code five times, employing the logistic loss function and Adam optimizer consistently. Specifically, during the training phase, each run is independently performed for 200 epochs with a batch size of 256. In balanced scenarios (*i.e.,*, $\pi = 0.5$), the learning rate is set to $10^{-3}$ across all datasets, with weight decay parameters set to $10^{-5}$ for MNIST, K-MNIST, F-MNIST, and CIFAR-10, $10^{-4}$ for Optdigits, and $10^{-3}$ for Pendigits. In imbalanced scenarios (*i.e.,*, $\pi = 0.2$), the learning rate is set to $10^{-4}$ for MNIST and K-MNIST, and $10^{-3}$ for the remaining datasets, with weight decay parameters set to $10^{-4}$ for K-MNIST and Optdigits, and $10^{-5}$ for the remaining datasets. During the pretraining phase, each run is independently executed for 20 epochs with a batch size of 256. The learning rate and weight decay remain consistent with those in the training phase.

### 4.2. Construction of the confidence differences

In this subsection, we present a method for generating confidence differences to validate the robustness of our method under noisy conditions.

**The confidence differences construction method**. The ConfDiff method generates class posterior probabilities using a logistic regression-based probabilistic classifier trained on labeled data and calculates the confidence difference according to its definition. Although this generation method facilitates comprehensive experimental analysis, it fails to accurately reflect the posterior probability distribution derived from manual annotations in real-world scenarios. To address this limitation, we propose an enhanced method that incorporates a posterior probability construction method based on outlier detection. This integration enables us to achieve a more uniform and realistic distribution while maintaining the fundamental definition of confidence differences. Our method consists of three main components: probability density estimation, outlier identification, and posterior probability rescaling.

First, we apply a Gaussian kernel-based probability density estimation method to the discrete posterior probabilities:

$$\hat{d}(\mathrm{x}_i) = \frac{1}{nh\sqrt{2\pi}} \sum_{j=1}^{n} \exp\left(-\frac{(\mathrm{x}_i - \mathrm{x}_j)^2}{2h^2}\right), \qquad (11)$$

where $\hat{d}(\mathrm{x}_i)$ represents the estimated probability density function at instance $\mathrm{x}_i$ and $\exp\left(-\frac{(\mathrm{x}_i - \mathrm{x}_j)^2}{2h^2}\right)$ is the standard Gaussian kernel function. The parameter $h$ denotes the kernel bandwidth controlling the degree of smoothing and is adaptively set based on the standard deviation of the probability distributions.

Next, we identify instances with densities below a threshold $o$ as outliers, where $o$ is also adaptively determined based on different probability density distributions. In our work, $o$ is set at the $2nd$ percentile of the probability density to avoid excessive filtering. For the remaining non-outlier instances, we rescale their posterior probabilities to ensure a more uniform distribution within the range $[0, 1]$:

$$p(y_i = +1|\mathbf{x}_i)$$
$$= \begin{cases} \text{Scaling}\left(p(y_i = +1|\mathbf{x}_i)\right), & \text{if } \hat{d}(\mathrm{x}_i) \leq o \\ p(y_i = +1|\mathbf{x}_i), & \text{otherwise} \end{cases}$$
$$\tag{12}$$

where $\text{Scaling}(\cdot)$ denotes a scaling function as:

$$\text{Scaling}(p_i) = \begin{cases} \log(p_i + \vartheta), & \text{if } p_i \leq 0.5 \\ \log(1 - p_i + \vartheta), & \text{otherwise} \end{cases} \tag{13}$$

where $\vartheta = e^{-10}$ is a smoothing parameter. Then, the confidence difference can be calculated according to its definition $c(\mathbf{x}_i, \mathbf{x}'_i) = p(y'_i = +1|\mathbf{x}'_i) - p(y_i = +1|\mathbf{x}_i)$.

**Artificially add noise.** One straightforward method is to add noise directly to $c$. However, it overlooks the intrinsic logic behind the original construction of $c$. To better simulate real-world conditions, we focus on observing how noise impacts the posterior probability distribution, thereby influencing $c$ indirectly. Specifically, we introduce noise to the posterior probabilities generated by the probabilistic classifier, which consequently adds noise to $c$. In the real world, individuals tend to exhibit smaller judgment biases towards more similar pairwise instances, while generating larger biases towards instances with lower similarity. Therefore, White Gaussian Noise (WGN) is introduced into the posterior probabilities $p(y_i = +1|\mathbf{x}_i)$ and $p(y'_i = +1|\mathbf{x}'_i)$ provided by the probabilistic classifier for the pairwise instance $(\mathbf{x}_i, \mathbf{x}'_i)$. Subsequently, the noisy posterior probabilities are used to generate the confidence difference, *i.e.,*,

*Table 2.* Classification accuracy of each comparing method on six datasets (mean±std) when $\pi = 0.5$, where the best performance is shown in boldface.

| $m$ | Method | MNIST | K-MNIST | F-MNIST | CIFAR-10 | Pendigits | Optdigits |
|---|---|---|---|---|---|---|---|
| 0 | Pcomp-Unbiased | 0.815±0.007 | 0.588±0.087 | 0.813±0.066 | 0.752±0.005 | **0.775±0.018** | 0.795±0.020 |
| | Pcomp-ReLU | 0.719±0.108 | 0.692±0.012 | 0.614±0.132 | 0.794±0.009 | 0.746±0.014 | 0.766±0.038 |
| | Pcomp-ABS | 0.830±0.005 | 0.727±0.015 | 0.837±0.010 | 0.828±0.006 | 0.645±0.059 | 0.722±0.027 |
| | Pcomp-Teacher | 0.882±0.024 | 0.708±0.008 | 0.887±0.012 | 0.812±0.010 | 0.496±0.016 | 0.507±0.067 |
| | ConfDiff-Unbiased | 0.723±0.072 | 0.576±0.029 | 0.771±0.085 | 0.848±0.014 | 0.675±0.071 | 0.799±0.023 |
| | ConfDiff-ReLU | 0.929±0.003 | 0.771±0.025 | 0.912±0.020 | 0.848±0.014 | 0.675±0.071 | 0.799±0.023 |
| | ConfDiff-ABS | 0.944±0.003 | 0.825±0.011 | 0.952±0.004 | 0.848±0.014 | 0.675±0.071 | 0.799±0.023 |
| | CRCR-Unbiased | 0.777±0.034 | 0.769±0.004 | 0.921±0.009 | **0.869±0.009** | 0.756±0.006 | **0.823±0.023** |
| | CRCR-ReLU | 0.919±0.019 | 0.685±0.080 | 0.925±0.017 | **0.869±0.009** | 0.753±0.007 | **0.823±0.023** |
| | CRCR-ABS | **0.962±0.006** | **0.848±0.013** | **0.955±0.002** | **0.869±0.009** | 0.753±0.009 | **0.823±0.023** |
| 50 | Pcomp-Unbiased | 0.814±0.050 | 0.606±0.086 | 0.855±0.061 | 0.733±0.010 | 0.760±0.020 | 0.793±0.022 |
| | Pcomp-ReLU | 0.849±0.008 | 0.722±0.003 | 0.833±0.063 | 0.810±0.008 | 0.756±0.036 | 0.772±0.017 |
| | Pcomp-ABS | 0.853±0.016 | 0.730±0.013 | 0.876±0.015 | 0.833±0.005 | 0.676±0.069 | 0.736±0.017 |
| | Pcomp-Teacher | 0.898±0.019 | 0.723±0.018 | 0.907±0.021 | 0.812±0.007 | 0.495±0.017 | 0.503±0.068 |
| | ConfDiff-Unbiased | 0.678±0.046 | 0.602±0.021 | 0.794±0.034 | 0.833±0.013 | 0.675±0.073 | 0.792±0.021 |
| | ConfDiff-ReLU | 0.933±0.002 | 0.766±0.020 | 0.933±0.012 | 0.836±0.014 | 0.675±0.073 | 0.792±0.021 |
| | ConfDiff-ABS | 0.937±0.004 | 0.819±0.007 | 0.953±0.007 | 0.834±0.013 | 0.675±0.073 | 0.792±0.021 |
| | CRCR-Unbiased | 0.845±0.043 | 0.779±0.008 | 0.928±0.001 | 0.859±0.003 | 0.759±0.029 | **0.821±0.022** |
| | CRCR-ReLU | 0.923±0.023 | 0.793±0.019 | 0.936±0.007 | **0.860±0.003** | 0.757±0.030 | **0.821±0.022** |
| | CRCR-ABS | **0.961±0.005** | **0.851±0.010** | **0.956±0.005** | **0.860±0.003** | 0.762±0.033 | **0.821±0.022** |
| 75 | Pcomp-Unbiased | 0.849±0.010 | 0.596±0.086 | 0.832±0.129 | 0.716±0.006 | 0.754±0.028 | 0.794±0.021 |
| | Pcomp-ReLU | 0.858±0.006 | 0.728±0.013 | 0.880±0.012 | 0.820±0.008 | 0.743±0.038 | 0.783±0.018 |
| | Pcomp-ABS | 0.865±0.008 | 0.734±0.017 | 0.874±0.011 | 0.836±0.003 | 0.688±0.060 | 0.743±0.020 |
| | Pcomp-Teacher | 0.908±0.010 | 0.735±0.013 | 0.920±0.018 | 0.813±0.008 | 0.495±0.018 | 0.501±0.069 |
| | ConfDiff-Unbiased | 0.620±0.084 | 0.560±0.025 | 0.650±0.051 | 0.844±0.008 | 0.674±0.073 | 0.795±0.018 |
| | ConfDiff-ReLU | 0.922±0.019 | 0.778±0.008 | 0.931±0.015 | 0.843±0.009 | 0.674±0.073 | 0.795±0.018 |
| | ConfDiff-ABS | 0.933±0.006 | 0.817±0.009 | 0.954±0.004 | 0.844±0.009 | 0.674±0.073 | 0.795±0.018 |
| | CRCR-Unbiased | 0.797±0.075 | 0.791±0.010 | 0.926±0.010 | **0.858±0.003** | 0.723±0.033 | **0.819±0.022** |
| | CRCR-ReLU | 0.938±0.006 | 0.792±0.010 | 0.942±0.005 | **0.858±0.003** | 0.721±0.035 | **0.819±0.022** |
| | CRCR-ABS | **0.962±0.003** | **0.851±0.006** | **0.959±0.001** | **0.858±0.003** | 0.756±0.009 | **0.819±0.022** |
| 100 | Pcomp-Unbiased | 0.832±0.051 | 0.631±0.079 | 0.897±0.013 | 0.708±0.014 | 0.735±0.024 | 0.796±0.015 |
| | Pcomp-ReLU | 0.862±0.015 | 0.726±0.012 | 0.883±0.017 | 0.827±0.004 | 0.725±0.035 | 0.787±0.019 |
| | Pcomp-ABS | 0.865±0.014 | 0.735±0.009 | 0.886±0.009 | 0.837±0.006 | 0.688±0.059 | 0.766±0.020 |
| | Pcomp-Teacher | 0.914±0.011 | 0.738±0.020 | 0.921±0.011 | 0.812±0.010 | 0.495±0.018 | 0.499±0.070 |
| | ConfDiff-Unbiased | 0.631±0.056 | 0.548±0.022 | 0.573±0.060 | 0.835±0.012 | 0.669±0.070 | 0.791±0.021 |
| | ConfDiff-ReLU | 0.920±0.014 | 0.769±0.008 | 0.923±0.032 | 0.834±0.012 | 0.669±0.070 | 0.791±0.021 |
| | ConfDiff-ABS | 0.934±0.006 | 0.812±0.004 | 0.953±0.005 | 0.835±0.012 | 0.669±0.070 | 0.791±0.021 |
| | CRCR-Unbiased | 0.860±0.081 | 0.804±0.009 | 0.910±0.030 | **0.851±0.007** | 0.751±0.008 | **0.815±0.019** |
| | CRCR-ReLU | 0.939±0.006 | 0.797±0.006 | 0.941±0.006 | **0.851±0.007** | **0.752±0.008** | **0.815±0.019** |
| | CRCR-ABS | **0.960±0.002** | **0.856±0.008** | **0.960±0.002** | **0.851±0.007** | **0.752±0.008** | **0.815±0.019** |

$\tilde{c}_i = \tilde{c}(\mathbf{x}_i, \mathbf{x}'_i) = \tilde{p}(y'_i = +1|\mathbf{x}'_i) - \tilde{p}(y_i = +1|\mathbf{x}_i)$, where

$$\tilde{p}(y'_i = +1|\mathbf{x}'_i) = p(y'_i = +1|\mathbf{x}'_i) + \zeta'_i, \quad \zeta'_i \sim N(0, \sigma^2)$$
$$\tilde{p}(y_i = +1|\mathbf{x}_i) = p(y_i = +1|\mathbf{x}_i) + \zeta_i, \quad \zeta_i \sim N(0, \sigma^2), \tag{14}$$

where $\zeta'_i$ and $\zeta_i$ represent the noise offsets which follow a standard Gaussian distribution $N(0, \sigma^2)$. In our experiments, we set $\sigma = 1/3$.

### 4.3. Result Analysis

Table 2 and Table 3 present the results of all baselines on four benchmark datasets and two UCI datasets for class-balanced (*i.e.,*, prior = 0.5) and class-imbalanced scenarios (*i.e.,*, prior = 0.2), respectively. Overall, our method performs nearly optimally across all scenarios, consistently achieving nearly the best results using the ABS risk correction function.

In scenarios with balanced classes, our method outperforms

Pcomp with accuracy improvements ranging from 0.02 to 0.341 across different datasets and surpasses ConfDiff with improvements ranging from 0.01 to 0.387, as observed from a baseline perspective. CRCR-ABS outperforms nearly all baselines, with the only observed exception being the results of Pcomp-Unbiased on the Pendigits dataset when no artificial noise is added. This exception may be due to the fact that the Pcomp method leverages only the information that one instance is more likely to be positive than another, without requiring knowledge of the exact difference between them. When no artificial noise is added, Pcomp's posterior probability reconstruction function preserves the monotonic increasing relationship of the posterior probabilities, without altering the relative likelihood of positivity between instances. Moreover, compared to Pcomp and ConfDiff, our method demonstrates increasingly stable and consistent accuracy as the noise ratio increases, with notable improvements in both accuracy and standard deviation, especially when the noise ratio reaches 100%. This results indicates its

*Table 3.* Classification accuracy of each comparing method on six datasets (mean±std) when $\pi = 0.2$, where the best performance is shown in boldface.

| $m$ | Method | MNIST | K-MNIST | F-MNIST | CIFAR-10 | Pendigits | Optdigits |
|---|---|---|---|---|---|---|---|
| 0 | Pcomp-Unbiased | 0.744±0.037 | 0.555±0.076 | 0.748±0.047 | 0.634±0.021 | 0.820±0.025 | 0.813±0.024 |
| | Pcomp-ReLU | 0.800±0.000 | 0.800±0.000 | 0.800±0.000 | 0.802±0.003 | 0.819±0.020 | 0.816±0.007 |
| | Pcomp-ABS | 0.804±0.009 | 0.800±0.000 | 0.801±0.001 | 0.833±0.004 | 0.797±0.023 | 0.805±0.006 |
| | Pcomp-Teacher | 0.788±0.074 | 0.695±0.046 | 0.883±0.026 | 0.813±0.020 | 0.482±0.212 | 0.684±0.097 |
| | ConfDiff-Unbiased | 0.743±0.033 | 0.622±0.077 | 0.724±0.025 | 0.812±0.004 | 0.797±0.028 | 0.830±0.016 |
| | ConfDiff-ReLU | 0.800±0.000 | 0.800±0.000 | 0.846±0.064 | 0.800±0.000 | 0.797±0.028 | 0.830±0.016 |
| | ConfDiff-ABS | 0.910±0.015 | 0.841±0.014 | **0.940±0.010** | 0.800±0.001 | 0.797±0.028 | 0.830±0.016 |
| | CRCR-Unbiased | 0.816±0.043 | 0.597±0.055 | 0.886±0.009 | **0.841±0.012** | **0.823±0.005** | **0.838±0.017** |
| | CRCR-ReLU | 0.929±0.055 | 0.814±0.031 | 0.930±0.049 | 0.801±0.001 | 0.817±0.012 | 0.830±0.008 |
| | CRCR-ABS | **0.916±0.022** | **0.856±0.006** | 0.922±0.007 | 0.812±0.017 | 0.784±0.024 | 0.825±0.005 |
| 50 | Pcomp-Unbiased | 0.742±0.015 | 0.547±0.038 | 0.768±0.070 | 0.623±0.017 | 0.818±0.025 | 0.810±0.027 |
| | Pcomp-ReLU | 0.800±0.000 | 0.801±0.002 | 0.800±0.000 | 0.801±0.003 | 0.806±0.023 | 0.821±0.007 |
| | Pcomp-ABS | 0.824±0.029 | 0.800±0.000 | 0.809±0.006 | 0.833±0.006 | 0.801±0.030 | 0.811±0.010 |
| | Pcomp-Teacher | 0.822±0.061 | 0.707±0.062 | 0.902±0.014 | 0.797+0.033 | 0.483±0.211 | 0.682±0.096 |
| | ConfDiff-Unbiased | 0.694±0.030 | 0.640±0.043 | 0.711±0.018 | 0.805±0.006 | 0.797±0.029 | 0.834±0.015 |
| | ConfDiff-ReLU | 0.800±0.000 | 0.800±0.000 | 0.821±0.046 | 0.800±0.001 | 0.797±0.029 | 0.834±0.015 |
| | ConfDiff-ABS | 0.891±0.025 | 0.818±0.010 | 0.938±0.014 | 0.801±0.002 | 0.797±0.029 | 0.834±0.015 |
| | CRCR-Unbiased | 0.794±0.043 | 0.623±0.079 | 0.880±0.016 | 0.789±0.035 | 0.795±0.025 | **0.843±0.023** |
| | CRCR-ReLU | 0.908±0.063 | 0.815±0.015 | 0.936±0.043 | 0.811±0.025 | 0.808±0.021 | 0.838±0.014 |
| | CRCR-ABS | **0.916±0.013** | **0.830±0.029** | **0.950±0.011** | **0.850±0.017** | **0.822±0.019** | 0.835±0.011 |
| 75 | Pcomp-Unbiased | 0.753±0.031 | 0.535±0.048 | 0.775±0.059 | 0.616±0.038 | 0.817±0.017 | 0.813±0.030 |
| | Pcomp-ReLU | 0.804±0.009 | 0.804±0.007 | 0.800±0.000 | 0.805±0.012 | 0.822±0.020 | 0.827±0.008 |
| | Pcomp-ABS | 0.863±0.014 | 0.800±0.000 | 0.828±0.016 | 0.832±0.005 | 0.803±0.038 | 0.813±0.009 |
| | Pcomp-Teacher | 0.840±0.061 | 0.714±0.055 | 0.908±0.019 | 0.793±0.044 | 0.482±0.211 | 0.680±0.096 |
| | ConfDiff-Unbiased | 0.704±0.058 | 0.630±0.026 | 0.745±0.088 | 0.804±0.004 | 0.796±0.031 | 0.830±0.016 |
| | ConfDiff-ReLU | 0.800±0.000 | 0.800±0.000 | 0.800±0.000 | 0.800±0.000 | 0.796±0.031 | 0.830±0.016 |
| | ConfDiff-ABS | 0.862±0.030 | 0.806±0.005 | 0.921±0.023 | 0.800±0.001 | 0.796±0.031 | 0.830±0.016 |
| | CRCR-Unbiased | 0.792±0.047 | 0.640±0.028 | 0.861±0.033 | 0.772±0.020 | 0.811±0.030 | 0.836±0.022 |
| | CRCR-ReLU | 0.901±0.055 | 0.817±0.024 | 0.801±0.001 | 0.828±0.024 | **0.827±0.008** | 0.836±0.017 |
| | CRCR-ABS | **0.914±0.008** | **0.819±0.022** | **0.947±0.006** | **0.853±0.004** | 0.819±0.011 | **0.839±0.012** |
| 100 | Pcomp-Unbiased | 0.752±0.021 | 0.540±0.069 | 0.834±0.034 | 0.643±0.053 | 0.805±0.024 | 0.817±0.027 |
| | Pcomp-ReLU | 0.845±0.040 | 0.808±0.010 | 0.814±0.019 | 0.806±0.005 | 0.808±0.020 | 0.834±0.008 |
| | Pcomp-ABS | 0.871±0.006 | 0.801±0.001 | 0.844±0.015 | 0.835±0.003 | 0.803±0.029 | 0.823±0.012 |
| | Pcomp-Teacher | 0.869±0.068 | 0.711±0.062 | 0.922±0.011 | 0.787±0.033 | 0.482±0.211 | 0.681±0.096 |
| | ConfDiff-Unbiased | 0.772±0.056 | 0.693±0.028 | 0.748±0.101 | 0.810±0.007 | 0.796±0.028 | 0.831±0.017 |
| | ConfDiff-ReLU | 0.800±0.000 | 0.800±0.000 | 0.800±0.000 | 0.800±0.001 | 0.796±0.028 | 0.831±0.016 |
| | ConfDiff-ABS | 0.814±0.006 | 0.801±0.002 | 0.870±0.043 | 0.801±0.001 | 0.796±0.028 | 0.831±0.016 |
| | CRCR-Unbiased | 0.790±0.036 | 0.639±0.059 | 0.838±0.056 | 0.780±0.014 | 0.797±0.018 | 0.837±0.026 |
| | CRCR-ReLU | 0.905±0.059 | 0.808±0.007 | 0.903±0.039 | 0.800±0.001 | 0.800±0.014 | 0.838±0.020 |
| | CRCR-ABS | **0.926±0.010** | **0.828±0.025** | **0.958±0.009** | **0.841±0.005** | **0.810±0.006** | **0.839±0.019** |

ability to produce more competitive results in the presence of artificial noise interference.

In scenarios with imbalanced classes, Pcomp-ReLU and ConfDiff-ReLU tend to exhibit random outcomes when confronted with imbalanced data augmented with artificial noise. This phenomenon may be attributed to the introduced noise, which significantly increases the likelihood of predictions where one instance in a pair is incorrectly predicted to be more likely positive than the other, contrary to the actual scenario. Such contradictions become significantly more pronounced as class imbalance and noise ratio increase. Compared to these methods, our method shows advantages in both accuracy mean and variance. From the dataset perspective, CRCR-ABS significantly outperforms other methods on the MNIST, K-MNIST, F-MNIST, and CIFAR-10 datasets in the presence of artificial noise, while maintaining strong competitiveness on the Pendigits and Optdigits datasets. CRCR-Unbiased shows promising results without artificial noise; however, the experiments

clearly demonstrate that its training challenges on complex and noisy datasets often lead to a notable decline in performance. This findings further underscore the effectiveness of CRCR-ABS in maintaining robust performance when dealing with complex datasets.

Such contradictions become significantly more pronounced as class imbalance and noise ratio increase. Compared to these methods, our approach shows advantages in both accuracy mean and variance. From the dataset perspective, CRCR_ABS significantly outperforms other methods on the MNIST, K-MNIST, F-MNIST, and CIFAR-10 datasets in the presence of artificial noise, while maintaining strong competitiveness on the Pendigits and Optdigits datasets. CRCR_Unbiased shows promising results without artificial noise; however, the experiments clearly demonstrate that its training challenges on complex and noisy datasets often lead to a notable decline in performance. These findings further underscore the effectiveness of CRCR_ABS in maintaining robust performance when dealing with complex datasets.

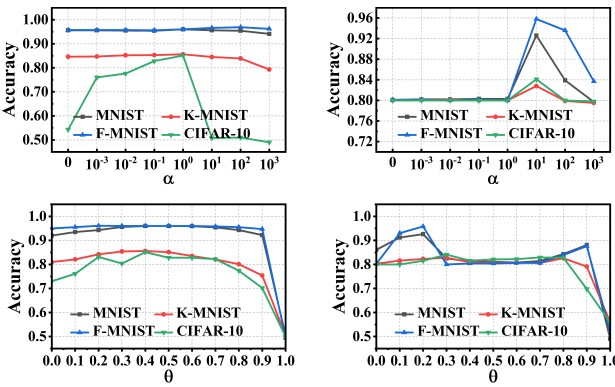

*Figure 2.* Sensitivity analysis of parameters $\alpha$ (top) and $\theta$ (bottom) on four benchmark datasets when $\pi = 0.5$ (left) and $\pi = 0.2$ (right).

### 4.4. Parameter Sensitivity

In this subsection, we conduct experiments with different thresholds $\theta$ for partitioning subsets and the parameter $\alpha$ for the consistency term, and the results are shown in Figure 2.

**About different threshold $\theta$** To evaluate the sensitivity of the threshold $\theta$, we vary its value within the range $\{0.1, 0.2, \ldots, 1\}$ on four distinct benchmark datasets (*i.e.,*, MNIST, K-MNIST, F-MNIST and CIFAR-10). The results reveal that the accuracy peaks for all datasets when $\theta = 0.4$ with $\pi = 0.5$, and when $\theta = 0.2$ or $0.3$ with $\pi = 0.2$. This observation may be attributed to the distribution of confidence differences resembling a normal distribution. A low threshold results in numerous inaccurate predictions within the subset $D^S$ utilized for risk consistency, while a high threshold leads to a scarcity of samples within $D^S$, thus diminishing the available supervisory information. Therefore, we empirically recommend setting the threshold at $\theta = 0.4$ when $\pi = 0.5$, and $\theta = \{0.2, 0.3\}$ when $\pi = 0.2$.

**About different parameter $\alpha$** To assess the sensitivity of the parameter $\alpha$, we vary its values across the range $\{10^i | i = -3, \ldots, +3\}$ on four benchmark datasets. Our analysis reveals that $\alpha$ shows increased sensitivity on the larger-scale CIFAR-10 dataset when $\pi = 0.5$, while maintaining relatively stable performance on the smaller-scale datasets. Moreover, $\alpha$ leads to a consistent trend in accuracy variation across the four datasets when $\pi = 0.2$. Notably, it achieves relatively optimal results when $\alpha = 1$ with $\pi = 0.5$, and $\alpha = 10$ with $\pi = 0.2$. Thus, we recommend setting $\alpha = 1$ or $10$ in experimental setups.

### 4.5. Ablation Study

In this subsection, we conduct ablation studies on various strategies by setting corresponding parameters to zero.

Specifically, setting $\alpha = 0$ and $\theta = 0$ represent versions without consistency strategy and without subset segmentation strategy, respectively. The experimental results, presented in Figure 2, demonstrate that our proposed subset segmentation strategy and consistency term contribute to performance improvement to some extent in the context of confidence difference classification under artificially added noise.

## 5. Conclusion

In this paper, we propose a novel ConfDiff classification method based on consistency risk and consistency regularization to address the challenge of noisy supervised signals in ConfDiff classification. We conduct a theoretical analysis of various supervised signals associated with different confidence differences. Based on this analysis, the ConfDiff dataset is partitioned into two subsets according to the reliability of the supervised information. For the subset with more reliable supervision, we employ consistency risk to preserve precise supervised information. Conversely, for the subset with less reliable supervision, we leverage consistency regularization to mitigate the impact of erroneous predictions. Extensive experimental results demonstrate that CRCR outperforms state-of-the-art baselines and exhibits strong robustness, even when artificial noise is introduced.

## Acknowledgements

We would like to acknowledge support for this project from the National Science and Technology Major Project (No.2021ZD0112500), and the National Natural Science Foundation of China (No.62276113).

## Impact Statement

The confidence difference classification proposed in this paper has the potential to significantly improve decision accuracy in real-world applications. It addresses potential noise impacts present in real-world data and holds substantial practical significance, especially in weakly supervised domains. This method is applicable to various fields, including medical diagnosis, rehabilitation assessment, and financial risk management.

However, it is important to acknowledge that the confidence differences used in our method may be affected by inherent data biases in real-world scenarios. Furthermore, while we demonstrate the effectiveness of our method in weakly supervised settings, there remains a risk of excessive dependence on algorithms for decision-making, which could potentially overlooking the cultivation of individual decision-making capabilities and autonomy.

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

# A. Proof of Theorem 3.1

In this appendix, we provide the proof of the Theorem 3.1 and the corresponding technical lemmas.

**Lemma A.1.** *The Rademacher complexity $\bar{\Re}_n(\mathcal{L}_{\mathrm{CRCR}} \circ \mathcal{G})$ on $\mathcal{D}$ for ConfDiff data with noise of size $n$ can be defined as follows:*

$$\bar{\Re}_n(\mathcal{L}_{\mathrm{CRCR}} \circ \mathcal{G}) \leq 2L_\ell \Re_{n_1}(\mathcal{G}) + \frac{\alpha}{\log(\varepsilon)} \Re_{n_2}(\mathcal{G}) \tag{15}$$

The proof of Lemma A.1:

$$
\begin{aligned}
\bar{\Re}_n(\mathcal{L}_{\mathrm{CRCR}} \circ \mathcal{G}) =& \mathbb{E}_{\mathcal{D}_{n_1}} \mathbb{E}_\sigma [\sup_{g \in \mathcal{G}} \frac{1}{n_1} \sum_{i=1}^{n_1} \sigma_i \mathcal{L}_{\mathrm{CRCR}}^S(g; \mathbf{x}_i, \mathbf{x}_i')] \\
& + \mathbb{E}_{\mathcal{D}_{n_2}} \mathbb{E}_\sigma [\sup_{g \in \mathcal{G}} \frac{1}{n_2} \sum_{i=1}^{n_2} \sigma_i \mathcal{L}_{\mathrm{CRCR}}^C(g; \mathbf{x}_i, \mathbf{x}_i')] \\
=& \mathbb{E}_{\mathcal{D}_{n_1}} \mathbb{E}_\sigma [\sup_{g \in \mathcal{G}} \frac{1}{n_1} \sum_{i=1}^{n_1} \frac{1}{2} \sigma_i ((\pi - c_i)\ell(g(\mathbf{x}_i), +1) + (1 - \pi - c_i)\ell(g(\mathbf{x}_i'), -1) \\
& \qquad\qquad\qquad + (\pi + c_i)\ell(g(\mathbf{x}_i'), +1) + (1 - \pi + c_i)\ell(g(\mathbf{x}_i), -1))] \\
& + \mathbb{E}_{\mathcal{D}_{n_2}} \mathbb{E}_\sigma [\sup_{g \in \mathcal{G}} \frac{1}{n_2} \sum_{i=1}^{n_2} \alpha \sigma_i \frac{1}{\log(|c_i| + \varepsilon)} \cdot \|(g(\mathbf{x}_i) - g(\mathbf{x}_i')\|_2] \\
=& \mathbb{E}_{\mathcal{D}_{n_1}} \mathbb{E}_\sigma [\sup_{g \in \mathcal{G}} \frac{1}{n_1} \sum_{i=1}^{n_1} \sigma_i \left\| \triangledown \mathcal{L}_{\mathrm{CRCR}}^S(g; \mathbf{x}_i, \mathbf{x}_i') \right\|_2 g(\mathbf{x}_i)] \\
& + \mathbb{E}_{\mathcal{D}_{n_2}} \mathbb{E}_\sigma [\sup_{g \in \mathcal{G}} \frac{1}{n_2} \sum_{i=1}^{n_2} \sigma_i \left\| \triangledown \mathcal{L}_{\mathrm{CRCR}}^C(g; \mathbf{x}_i, \mathbf{x}_i') \right\|_2 g(\mathbf{x}_i)]
\end{aligned}
\tag{16}
$$

where

$$
\begin{aligned}
& \left\| \triangledown \mathcal{L}_{\mathrm{CRCR}}^S(g; \mathbf{x}_i, \mathbf{x}_i') \right\|_2 \\
=& \frac{1}{2} \left\| \triangledown ((\pi - c_i)\ell(g(\mathbf{x}_i), +1) + (1 - \pi - c_i)\ell(g(\mathbf{x}_i'), -1) \right. \\
& \left. + (\pi + c_i)\ell(g(\mathbf{x}_i'), +1) + (1 - \pi + c_i)\ell(g(\mathbf{x}_i), -1)) \right\|_2 \\
\leq& \frac{1}{2} \Big( \left\| \triangledown ((\pi - c_i)\ell(g(\mathbf{x}_i), +1)) \right\|_2 + \left\| \triangledown ((1 - \pi - c_i)\ell(g(\mathbf{x}_i'), -1)) \right\|_2 \\
& + \left\| \triangledown ((\pi + c_i)\ell(g(\mathbf{x}_i'), +1)) \right\|_2 + \left\| \triangledown ((1 - \pi + c_i)\ell(g(\mathbf{x}_i), -1)) \right\|_2 \Big) \\
\leq& \frac{1}{2} |\pi - c_i| L_\ell + \frac{1}{2} |1 - \pi - c_i| L_\ell + \frac{1}{2} |\pi + c_i| L_\ell + \frac{1}{2} |1 - \pi + c_i| L_\ell \\
\leq& 2L_\ell
\end{aligned}
\tag{17}
$$
$$\tag{18}$$

and,

$$
\begin{aligned}
\left\| \triangledown \mathcal{L}_{\mathrm{CRCR}}^C(g; \mathbf{x}_i, \mathbf{x}_i') \right\|_2 =& \alpha \left\| \triangledown \frac{1}{\log(|c_i| + \varepsilon)} \cdot \|(g(\mathbf{x}_i) - g(\mathbf{x}_i')\|_2 \right\|_2 \\
\leq& \alpha \frac{1}{\log(|c_i| + \varepsilon)} \cdot \frac{g(\mathbf{x}_i) - g(\mathbf{x}_i')}{\|g(\mathbf{x}_i) - g(\mathbf{x}_i')\|_2} \\
\leq& \frac{\alpha}{\log(\varepsilon)}
\end{aligned}
\tag{19}
$$

Replacing the corresponding term in Eq.16 with Eq.18 and Eq.19, we can prove the Lemma A.1:

$$\bar{\Re}_n(\mathcal{L}_{\text{CRCR}} \circ \mathcal{G}) \leq 2L_\ell \mathbb{E}_{\mathcal{D}_{n_1}} \mathbb{E}_\sigma [\sup_{g \in \mathcal{G}} \frac{1}{n_1} \sum_{i=1}^{n_1} \sigma_i g(\mathbf{x}_i)] + \frac{\alpha}{\log(\varepsilon)} \mathbb{E}_{\mathcal{D}_{n_2}} \mathbb{E}_\sigma [\sup_{g \in \mathcal{G}} \frac{1}{n_2} \sum_{i=1}^{n_2} \sigma_i g(\mathbf{x}_i)]$$

$$\leq 2L_\ell \Re_{n_1}(\mathcal{G}) + \frac{\alpha}{\log(\varepsilon)} \Re_{n_2}(\mathcal{G}) \tag{20}$$

**Lemma A.2.**

$$\sup_{g \in \mathcal{G}} \left| R(g) - \hat{R}_{\text{CRCR}}(g) \right| \leq (4 + 2\beta)L_\ell \Re_{n_1}(\mathcal{G}) + \frac{3\alpha}{\log(\varepsilon)} \Re_{n_2}(\mathcal{G}) + C_\ell \sqrt{\frac{\ln(2/\delta)}{2n_1}} \tag{21}$$

The proof of Lemma A.2:

$$\sup_{g \in \mathcal{G}} \left| R(g) - \hat{R}_{\text{CRCR}}(g) \right| \leq \sup_{g \in \mathcal{G}} \left| R(g) - \mathbb{E}[\hat{R}_{\text{CRCR}}(g)] \right| + \sup_{g \in \mathcal{G}} \left| \mathbb{E}[\hat{R}_{\text{CRCR}}(g)] - \hat{R}_{\text{CRCR}}(g) \right| \tag{22}$$

We first discuss the first term (the bias term). Since $R_{\text{CD}}(g)$ is an unbiased risk estimator of $R(g)$, we have:

$$\left| R(g) - \mathbb{E}[\hat{R}_{\text{CRCR}}(g)] \right| = \left| R(g) - R_{\text{CD}}(g) + R_{\text{CD}}(g) - \mathbb{E}[\hat{R}_{\text{CRCR}}(g)] \right|$$

$$\leq |R(g) - R_{\text{CD}}(g)| + \left| R_{\text{CD}}(g) - \mathbb{E}[\hat{R}_{\text{CRCR}}(g)] \right|$$

$$= \left| R_{\text{CD}}(g) - \mathbb{E}[\hat{R}_{\text{CRCR}}(g)] \right|$$

$$\leq \left| R_{\text{CD}}(g; D_S) - \mathbb{E}[\hat{R}_{\text{CD}}(g; D_S)] \right| + \alpha \left| \cdot \mathbb{E}_{D_C} \left[ \widehat{\text{Reg}}(g) \right] \right| \tag{23}$$

where $\widehat{\text{Reg}}(g) = \frac{\alpha}{n_2} \sum_{i=1}^{n_2} \left( \frac{1}{\log(|c_i| + \varepsilon)} \cdot \|g(\mathbf{x}_i) - g(\mathbf{x}_i')\|_2 \right)$ is the regularization term.

Let $p(\mathbf{x}, \mathbf{x}')$ denote the original distribution, and $p_{\mathcal{D}^S}(\mathbf{x}, \mathbf{x}')$ denote the conditional distribution of $\mathcal{D}^S$. We can define the density ratio as: $w(\mathbf{x}, \mathbf{x}') = \frac{p(\mathbf{x}, \mathbf{x}')}{p_{\mathcal{D}^S}(\mathbf{x}, \mathbf{x}')}$. Assuming there exists a constant $\beta > 0$ such that $w(\mathbf{x}, \mathbf{x}') = \frac{p(\mathbf{x}, \mathbf{x}')}{p_{\mathcal{D}^S}(\mathbf{x}, \mathbf{x}')} \leq \beta$ for all $(\mathbf{x}, \mathbf{x}') \in \mathcal{D}^S$. Under this assumption, following the theoretical framework proposed by (Cortes et al., 2010), we can derive the following generalization bound for the weighted ConfDiff risk estimation error:

$$\sup_{g \in \mathcal{G}} \left| R_{\text{CD}}(g) - \mathbb{E}[\hat{R}_{\text{CD}}(g; D_S)] \right| \leq 2\beta \Re_{n_1}(\mathcal{G}) + C_\ell \sqrt{\frac{\ln(2/\delta)}{2n_1}} \tag{24}$$

For the regularization term $\left| \mathbb{E}_{D_C} \left[ \widehat{\text{Reg}}(g) \right] \right|$, we have:

$$\sup_{g \in \mathcal{G}} \left| \mathbb{E}_{D_C} \left[ \widehat{\text{Reg}}(g) \right] \right| \leq \frac{1}{\log(\varepsilon)} \Re_{n_2}(\mathcal{G}) \tag{25}$$

In summary, the bias term satisfies:

$$\left| R(g) - \mathbb{E}[\hat{R}_{\text{CRCR}}(g)] \right| \leq \left| R_{\text{CD}}(g) - \mathbb{E}[\hat{R}_{\text{CD}}(g; D_S)] \right| + \alpha \left| \mathbb{E}_{D_C} \left[ \widehat{\text{Reg}}(g) \right] \right|$$

$$\leq 2\beta L_\ell \Re_{n_1}(\mathcal{G}) + C_\ell \sqrt{\frac{\ln(2/\delta)}{2n_1}} + \frac{\alpha}{\log(\varepsilon)} \Re_{n_2}(\mathcal{G}) \tag{26}$$

Then we discuss the second term. Since the quantity $\sup_{g \in \mathcal{G}} \left| \mathbb{E}[\hat{R}_{\text{CRCR}}(g)] - \hat{R}_{\text{CRCR}}(g) \right|$ is difficult to handle directly, let $S, S'$ are two independent samples of the same size (Mohri et al., 2012), we upper bound it by:

$$
\begin{aligned}
\sup_{g \in \mathcal{G}} \left| \mathbb{E}[\hat{R}_{\text{CRCR}}(g)] - \hat{R}_{\text{CRCR}}(g) \right| &\leq \mathbb{E}_S \left[ \sup_{g \in \mathcal{G}} \left( \mathbb{E}_{S'}[\hat{R}_{\text{CRCR},S'}(g)] - \hat{R}_{\text{CRCR},S}(g) \right) \right] \\
&= \mathbb{E}_{S,S'} \left[ \sup_{g \in \mathcal{G}} \left( \hat{R}_{\text{CRCR},S'}(g) - \hat{R}_{\text{CRCR},S}(g) \right) \right] \\
&= \mathbb{E}_{S,S'} \left[ \sup_{g \in \mathcal{G}} \frac{1}{n} \sum_{i=1}^{n} \left( \ell_{\text{CRCR}}(g, z_i') - \ell_{\text{CRCR}}(g, z_i) \right) \right] \\
&= \mathbb{E}_{S,S',\sigma} \left[ \sup_{g \in \mathcal{G}} \frac{1}{n} \sum_{i=1}^{n} \sigma_i \left( \ell_{\text{CRCR}}(g, z_i') - \ell_{\text{CRCR}}(g, z_i) \right) \right] \\
&\leq \mathbb{E}_{S',\sigma} \left[ \sup_{g \in \mathcal{G}} \frac{1}{n} \sum_{i=1}^{n} \sigma_i \ell_{\text{CRCR}}(g, z_i') \right] + \mathbb{E}_{S,\sigma} \left[ \sup_{g \in \mathcal{G}} \frac{1}{n} \sum_{i=1}^{n} (-\sigma_i) \ell_{\text{CRCR}}(g, z_i) \right] \\
&= 2\mathbb{E}_S \mathbb{E}_\sigma \left[ \sup_{g \in \mathcal{G}} \frac{1}{n} \sum_{i=1}^{n} \sigma_i \ell_{\text{CRCR}}(g, z_i) \right] \\
&= 2\bar{\Re}_n(\mathcal{L}_{\text{CRCR}} \circ \mathcal{G}) \\
&\leq 4L_\ell \Re_{n_1}(\mathcal{G}) + \frac{2\alpha}{\log(\varepsilon)} \Re_{n_2}(\mathcal{G}) \tag{27}
\end{aligned}
$$

Then the upper bound of the Lemma A.2 can be expressed as:

$$
\begin{aligned}
\sup_{g \in \mathcal{G}} \left| R(g) - \hat{R}_{\text{CRCR}}(g) \right| &\leq \sup_{g \in \mathcal{G}} \left| R(g) - \mathbb{E}[\hat{R}_{\text{CRCR}}(g)] \right| + \sup_{g \in \mathcal{G}} \left| \mathbb{E}[\hat{R}_{\text{CRCR}}(g)] - \hat{R}_{\text{CRCR}}(g) \right| \\
&\leq 2\beta L_\ell \Re_{n_1}(\mathcal{G}) + C_\ell \sqrt{\frac{\ln(2/\delta)}{2n_1}} + \frac{\alpha}{\log(\varepsilon)} \Re_{n_2}(\mathcal{G}) + 4L_\ell \Re_{n_1}(\mathcal{G}) + \frac{2\alpha}{\log(\varepsilon)} \Re_{n_2}(\mathcal{G}) \\
&= (4 + 2\beta) L_\ell \Re_{n_1}(\mathcal{G}) + \frac{3\alpha}{\log(\varepsilon)} \Re_{n_2}(\mathcal{G}) + C_\ell \sqrt{\frac{\ln(2/\delta)}{2n_1}} \tag{28}
\end{aligned}
$$

The proof of Theorem 3.1:

$$
\begin{aligned}
R(\hat{g}_{\text{CRCR}}) - R(g^*) &= \left( R(\hat{g}_{\text{CRCR}}) - \hat{R}_{\text{CRCR}}(\hat{g}_{\text{CRCR}}) \right) + \left( \hat{R}_{\text{CRCR}}(\hat{g}_{\text{CRCR}}) - \hat{R}_{\text{CRCR}}(g^*) \right) \\
&\quad + \left( \hat{R}_{\text{CRCR}}(g^*) - R(g^*) \right) \\
&\leq \left( R(\hat{g}_{\text{CRCR}}) - \hat{R}_{\text{CRCR}}(\hat{g}_{\text{CRCR}}) \right) + \left( \hat{R}_{\text{CRCR}}(g^*) - R(g^*) \right) \\
&\leq 2 \sup_{g \in \mathcal{G}} \left| R(g) - \hat{R}_{\text{CRCR}}(g) \right| \\
&\leq (8 + 4\beta) L_\ell \Re_{n_1}(\mathcal{G}) + \frac{6\alpha}{\log(\varepsilon)} \Re_{n_2}(\mathcal{G}) + 2C_\ell \sqrt{\frac{\ln(2/\delta)}{2n_1}} \tag{29}
\end{aligned}
$$

## B. Proof of Eq. 6

In this appendix, we provide the proof of the Eq. 6.

Substituting the form of the loss function from Eq.5 into Eq.3, then we can obtain:

$$
\begin{aligned}
R_{\mathrm{CD}}(g) &= \frac{1}{2}\mathbb{E}_{p(\mathbf{x},\mathbf{x}')}\Big[\big(\pi - c(\mathbf{x},\mathbf{x}')\big)\ell\big(g(\mathbf{x}),+1\big) + \big(1 - \pi - c(\mathbf{x},\mathbf{x}')\big)\ell\big(g(\mathbf{x}'),-1\big) \\
&\qquad + \big(\pi + c(\mathbf{x},\mathbf{x}')\big)\ell\big(g(\mathbf{x}'),+1\big) + \big(1 - \pi + c(\mathbf{x},\mathbf{x}')\big)\ell\big(g(\mathbf{x}),-1\big)\Big] \\
&= \frac{1}{2}\mathbb{E}_{p(\mathbf{x},\mathbf{x}')}\Big[\big(\pi - c(\mathbf{x},\mathbf{x}')\big)\big(h(g(\mathbf{x})) - g(\mathbf{x})\big) + \big(1 - \pi - c(\mathbf{x},\mathbf{x}')\big)\big(h(g(\mathbf{x}')) + g(\mathbf{x}')\big) \\
&\qquad + \big(\pi + c(\mathbf{x},\mathbf{x}')\big)\big(h(g(\mathbf{x}')) - g(\mathbf{x}')\big) + \big(1 - \pi + c(\mathbf{x},\mathbf{x}')\big)\big(h(g(\mathbf{x})) + g(\mathbf{x})\big)\Big] \\
&= \frac{1}{2}\mathbb{E}_{p(\mathbf{x},\mathbf{x}')}\Big[h\big(g(\mathbf{x})\big) + \big(1 - 2\pi + 2c(\mathbf{x},\mathbf{x}')\big)g(\mathbf{x}) \\
&\qquad + h\big(g(\mathbf{x}')\big) + \big(1 - 2\pi - 2c(\mathbf{x},\mathbf{x}')\big)g(\mathbf{x}')\Big] \\
&= \frac{1}{2}\mathbb{E}_{p(\mathbf{x},\mathbf{x}')}\Big[h\big(g(\mathbf{x})\big) + 2c(\mathbf{x},\mathbf{x}')g(\mathbf{x}) + h\big(g(\mathbf{x}')\big) - 2c(\mathbf{x},\mathbf{x}')g(\mathbf{x}')\Big] \\
&\qquad + \frac{1}{2}\mathbb{E}_{p(\mathbf{x},\mathbf{x}')}\Big[\big(1 - 2\pi\big)\big(g(\mathbf{x}) + g(\mathbf{x}')\big)\Big] \\
&= \frac{1}{2}\mathbb{E}_{p(\mathbf{x},\mathbf{x}')}\Big[h\big(g(\mathbf{x})\big) + 2c(\mathbf{x},\mathbf{x}')g(\mathbf{x}) + h\big(g(\mathbf{x}')\big) - 2c(\mathbf{x},\mathbf{x}')g(\mathbf{x}') \\
&\qquad + c(\mathbf{x},\mathbf{x}')h\big(g(\mathbf{x})\big) - c(\mathbf{x},\mathbf{x}')h\big(g(\mathbf{x})\big) + \frac{1}{2}g(\mathbf{x}) - \frac{1}{2}g(\mathbf{x}) \\
&\qquad + c(\mathbf{x},\mathbf{x}')h\big(g(\mathbf{x}')\big) - c(\mathbf{x},\mathbf{x}')h\big(g(\mathbf{x}')\big) + \frac{1}{2}g(\mathbf{x}') - \frac{1}{2}g(\mathbf{x}')\Big] \\
&\qquad + \frac{1}{2}\mathbb{E}_{p(\mathbf{x},\mathbf{x}')}\Big[\big(1 - 2\pi\big)\big(g(\mathbf{x}) + g(\mathbf{x}')\big)\Big] \\
&= \frac{1}{2}\mathbb{E}_{p(\mathbf{x},\mathbf{x}')}\Big[\frac{1}{2}h\big(g(\mathbf{x})\big) - c(\mathbf{x},\mathbf{x}')h\big(g(\mathbf{x})\big) - \frac{1}{2}g(\mathbf{x}) + c(\mathbf{x},\mathbf{x}')g(\mathbf{x}) \\
&\qquad + \frac{1}{2}h\big(g(\mathbf{x}')\big) + c(\mathbf{x},\mathbf{x}')h\big(g(\mathbf{x}')\big) - \frac{1}{2}g(\mathbf{x}') - c(\mathbf{x},\mathbf{x}')g(\mathbf{x}')\Big] \\
&\qquad + \frac{1}{2}\mathbb{E}_{p(\mathbf{x},\mathbf{x}')}\Big[\big(1 - 2\pi\big)\big(g(\mathbf{x}) + g(\mathbf{x}')\big)\Big] \\
&= \frac{1}{2}\mathbb{E}_{p(\mathbf{x},\mathbf{x}')}\Big[\frac{1}{2}h\big(g(\mathbf{x})\big) - c(\mathbf{x},\mathbf{x}')h\big(g(\mathbf{x})\big) - \frac{1}{2}g(\mathbf{x}) + c(\mathbf{x},\mathbf{x}')g(\mathbf{x}) \\
&\qquad + \frac{1}{2}h\big(g(\mathbf{x}')\big) + c(\mathbf{x},\mathbf{x}')h\big(g(\mathbf{x}')\big) - \frac{1}{2}g(\mathbf{x}') - c(\mathbf{x},\mathbf{x}')g(\mathbf{x}')\Big] \\
&\qquad + \frac{1}{2}\mathbb{E}_{p(\mathbf{x},\mathbf{x}')}\Big[\frac{1}{2}h\big(g(\mathbf{x})\big) + c(\mathbf{x},\mathbf{x}')h\big(g(\mathbf{x})\big) + \frac{1}{2}g(\mathbf{x}) + c(\mathbf{x},\mathbf{x}')g(\mathbf{x}) \\
&\qquad + \frac{1}{2}h\big(g(\mathbf{x}')\big) - c(\mathbf{x},\mathbf{x}')h\big(g(\mathbf{x}')\big) + \frac{1}{2}g(\mathbf{x}') - c(\mathbf{x},\mathbf{x}')g(\mathbf{x}')\Big] \\
&\qquad + \frac{1}{2}\mathbb{E}_{p(\mathbf{x},\mathbf{x}')}\Big[\big(1 - 2\pi\big)\big(g(\mathbf{x}) + g(\mathbf{x}')\big)\Big] \\
&= \frac{1}{2}\mathbb{E}_{p(\mathbf{x},\mathbf{x}')}\Big[\big(\tfrac{1}{2} - c(\mathbf{x},\mathbf{x}')\big)\ell\big(g(\mathbf{x}),+1\big) + \big(\tfrac{1}{2} + c(\mathbf{x},\mathbf{x}')\big)\ell\big(g(\mathbf{x}'),+1\big)\Big] \\
&\qquad + \frac{1}{2}\mathbb{E}_{p(\mathbf{x},\mathbf{x}')}\Big[\big(\tfrac{1}{2} + c(\mathbf{x},\mathbf{x}')\big)\ell\big(g(\mathbf{x}),-1\big) + \big(\tfrac{1}{2} - c(\mathbf{x},\mathbf{x}')\big)\ell\big(g(\mathbf{x}'),-1\big)\Big] \\
&\qquad + \frac{1}{2}\mathbb{E}_{p(\mathbf{x},\mathbf{x}')}\Big[\big(1 - 2\pi\big)\big(g(\mathbf{x}) + g(\mathbf{x}')\big)\Big]. \tag{30}
\end{aligned}
$$

Then Eq. 6 is proven.

