# OpenReview forum: "Confidence Difference Reflects Various Supervised Signals in Confidence-Difference Classification"
_ICML.cc/2025/Conference — ICML 2025 poster_

### Official Review · Reviewer_B2Kr · 2025-02-19

**Overall Recommendation:** 5

**Summary:**

This paper deals with confidence-difference classification, a weakly supervised binary classification problem. To mitigate the noise contained in the confidence differences, a novel risk estimator using consistency regularization is employed to improve performance. Extensive experiments on benchmark datasets validate the effectiveness of the proposed method.

**Claims And Evidence:**

The claim that different examples have different monitoring signals is novel and valid. Empirical analyses show that examples with $c>0.5$ carry more information, while examples with $c<0.5$ should be considered separately.

**Essential References Not Discussed:**

All the essential references are discussed.

**Experimental Designs Or Analyses:**

The experimental designs are good. First, the experimental data sets are comprehensive and the methods compared are current. Second, the experimental analyses are good. Third, ablation studies and sensitivity analyses are performed.

**Methods And Evaluation Criteria:**

The proposed methods are reasonable and the effectiveness is validated by both theoretical analysis and experimental results. The evaluation criteria are reasonable and follow commonly used protocols in the literature.

**Other Comments Or Suggestions:**

- There are some notations that need to be revised. For example, line 183 should read $g$ instead of $G$. The condition in line 184 is also incorrect. The details should be checked carefully.

**Other Strengths And Weaknesses:**

### Strengths
- The problem studied is novel in the literature. It is natural that different confidence differences carry different supervision signals in ConfDiff classification. A simple and effective regularization term is added to the previous unbiased risk estimator, and good performance is achieved.

- The paper is generally well written.

- The effectiveness of the proposed method is supported by solid theoretical analysis and extensive experiments.

### Weakenesses
- Line 161 says that if $c>0.5$, the two examples belong to different classes. This is not true, because the label is not related to the posterior probability. For example, negative data can have a large true posterior probability ($p=0.8$) and positive data can have a small true posterior probability ($p=0.2$). Therefore, the two examples can still belong to the same class.

- I am not sure whether equation (7) is biased from the original classification risk in equation (1). This is because the marginal distribution may change after data partitioning. If so, will this affect the theoretical analysis in Theorem 3.1, since the minimizers of the two risks are not the same?

**Questions For Authors:**

Please see "Weaknesses".

**Relation To Broader Scientific Literature:**

N/A.

**Theoretical Claims:**

The theoretical claims are correct.

---

> ### Author Rebuttal · Authors · 2025-03-31
>
> **Q1. If $c>0.5$, the two examples belong to different classes. This is not true.**
>
> Thank you for the insightful correction. The statement "$c>0.5$, the two examples belong to different classes" is indeed not precise enough. A more accurate expression would be "$c > 0.5$, the two examples belong to different classes **at a high probability**". This is also consistent with the data partitioning strategy adopted in our paper, which separates the dataset into a subset $D^S$ with relatively precise predictive information (i.e., $c > 0.5$) and a subset $D^C$ with comparatively imprecise predictive information (i.e., $c \le 0.5$).
> Moreover, we agree with your point that, there is no direct correspondence between the labels and the posterior probability during training. Our goal is to enable the model to learn and establish this relationship through training.
>
> We will revise this statement to be more rigorous in the next version.
>
> &nbsp;
>
> **Q2. I am not sure whether equation (7) is biased from the original classification risk in equation (1). This is because the marginal distribution may change after data partitioning. If so, will this affect the theoretical analysis in Theorem 3.1, since the minimizers of the two risks are not the same?**
>
> Thank you for your comments. We agree that Eq.7 constitutes a biased estimator of the original classification risk in Eq.1, due to the change in the marginal distribution induced by the subset partitioning strategy. However, this bias does not affect the theoretical analysis in Theorem 3.1.
>
> In Theorem 3.1, we explicitly model the risks over the two subsets separately, and analyze their contributions through the Rademacher complexities $\mathfrak{R}\_{n_1}(\mathcal{G})$ and $\mathfrak{R}\_{n_2}(\mathcal{G})$, respectively. The first and second terms of the error bound directly involve these complexities. So as $n_1, n_2 \to \infty$, both $\mathfrak{R}\_{n_1}(\mathcal{G})$ and $\mathfrak{R}\_{n_2}(\mathcal{G})$ tend to zero. Additionally, the third term, which depends on $\sqrt{n}/n_1$ and $\sqrt{n}/n_2$, also diminishes as the sample sizes increase. Consequently, as long as $n_1, n_2 \to \infty$, the overall estimation error still converges to the minimum of the original classification risk $R(g^*)$.
>
> &nbsp;
>
> **Q3. Line 183 should read instead $g$ of $\mathcal{G} $. The condition in line 184 is also incorrect.**
>
> Thanks for your correction. We will revise them in the next version.

---

> > ### Comment · Reviewer_B2Kr · 2025-04-02
> >
> > Thanks for the rebuttal. My concern has been addressed and I will increase my score to vote for acceptance on this paper.

---

### Official Review · Reviewer_aabi · 2025-03-13

**Overall Recommendation:** 3

**Summary:**

In this paper, the authors identify that noise supervision signals emerge in current confidence difference classification methods when the confidence difference is small. Based on this observation, the core focus of this work is to explore a robust solution for confidence difference classification by mitigating the impact of inaccurate supervised signals. This paper proposes a novel robust confidence difference classification method, constructing a risk estimator based on consistency risk and consistency regularization (CRCR), and theoretically derives the error bound of CRCR. The idea presented in this paper is interesting, and extensive experiments on diverse datasets under artificial noise interventions support the generalization capability of the proposed method.

**Claims And Evidence:**

The claims presented in this paper are supported by clear and convincing evidence. Specifically, this paper introduces a new risk estimator by analyzing the various supervised signals reflected in the ConfDiff method. The proposed risk estimator has been empirically validated; its error bound has been theoretically analyzed; and its robustness has been further verified under artificial noise intervention.

**Essential References Not Discussed:**

I don't see any essential related works missing from the citations.

**Experimental Designs Or Analyses:**

I've rigorously reviewed the experimental designs and analyses. In addition to conventional experimental designs, this paper introduces artificial noise at different levels and proposes a method to generate artificial noise to simulate potential real-world noise distributions. This's an interesting perspective.

**Methods And Evaluation Criteria:**

The proposed method, CRCR, effectively addresses the issue of noisy supervised signals in the ConfDiff method. These noisy signals tend to encourage the classifier to make predictions in the opposite direction, making this a novel and interesting research direction.

**Other Comments Or Suggestions:**

Please refer to the Weaknesses and Questions.

**Other Strengths And Weaknesses:**

Strengths:
1) This paper is motivated by the noisy supervised signals introduced in the ConfDiff method. The proposed method, CRCR, effectively addresses this issue, making it a novel and interesting research direction.
2) The proposed method, which constructs a risk estimator based on consistency risk and consistency regularization, is effective and is supported by both theoretical analysis and experimental validation.
3) In addition to traditional experimental settings, this paper designs an artificial noise generation method for confidence difference classification. The goal is to test whether the proposed method can maintain robustness under different levels of artificial noise interventions. The experimental results confirm this robustness.

Weaknesses:
1) Some details are not well explained, as noted in the Questions section.
2) The reasoning behind the overfitting issue should be better explained.
3) This paper follows a style similar to the ConfDiff method. However, the introduction of artificial noise at different levels is a noteworthy and distinctive highlight compared to prior work.

**Questions For Authors:**

1) Eq.5 provides a general form of many commonly used loss functions. Which specific loss functions are included? Is the logistic loss function used in the code also part of this general form?
2) Figure 1 is somewhat difficult to understand. Could you provide a clearer explanation of the meaning of the x-axis in Figure 1?
3) What is the purpose of the design in Section 3.4? Why might negative empirical risk lead to severe overfitting? And how does the risk correction function address this issue? This paper seems to lack a reasonable explanation.
4) The consistency regularization term encourages consistency between confidence differences and model outputs. Would this lead to more instance pairs with smaller confidence differences, thereby increasing the presence of noisy supervised signals?

**Relation To Broader Scientific Literature:**

The main motivation of this paper is an observation by analyzing the various supervised signals in ConfDiff method from both experimental and theoretical perspectives. The authors found that the ConfDiff method introduces noisy supervised signals when the confidence difference is small. Consequently, this paper focuses on mitigating the challenges posed by these noisy supervised signals.

**Theoretical Claims:**

I've carefully checked the correctness of the theoretical claims presented in this paper.
1) This paper analyzes the various supervised signals reflected by different confidence differences in ConfDiff classification from both experimental and theoretical perspectives, supporting the claim that noisy supervised signals exist in the ConfDiff method.
2) This paper also derives the error bound of the proposed method and provides detailed proofs and theoretical analysis in the appendix.

---

> ### Author Rebuttal · Authors · 2025-03-31
>
> **Q1. Which specific loss functions (e.g. logistic) are included in Eq.5?**
>
> Thank you for your comments. This function class includes several commonly used loss functions, such as those derived from Generalized Linear Models (GLMs), including the mean squared error (MSE) for linear regression, the logistic loss for logistic regression, the Poisson loss for Poisson regression, the exponential loss for exponential regression, and the cross-entropy loss for neural networks (Zhang et al., 2021).
>
> Zhang, L., Deng, Z., Kawaguchi, K., Ghorbani, A., and Zou, J. How does mixup help with robustness and generalization? In International Conference on Learning Representations, 2021.
>
> &nbsp;
>
> **Q2. Could you provide a clearer explanation of the x-axis in Fig.1?**
>
> Thank you for your suggestions. The x-axis values multiplied by $\pi$ represent the proportion of pairwise instances $(\mathbf{x}, \mathbf{x}')$ with confidence differences $c(\mathbf{x}, \mathbf{x}') \in [-1, -0.5) \cup (0.5, 1]$ relative to all pairwise instances. Thus, the x-axis values effectively serve as a scaling factor used for computing the proportion.
>
> &nbsp;
>
> **Q3. What is the purpose of the design in Section 3.4? Why might negative empirical risk lead to severe overfitting? And how does the risk correction function address this issue?**
>
> Many thanks for your comment. The purpose of Section 3.4 is to address the overfitting problem when using flexible models due to negative empirical risk.
>
> Risk is typically non-negative, reflecting the deviation between model predictions and ground-truth values. The objective of optimization is to minimize risk; if risk could be negative, the model could be inclined to find an optimization direction that continually reduces risk on the training data, leading to overfitting by learning noise and performing poorly on test data. Additionally, (Lu et al., 2020) highlights that negative empirical risk may be a potential cause of overfitting and experimentally demonstrates a strong co-occurrence of negative risk and overfitting across various models and datasets.
>
> The risk correction function enforces the non-negativity of the risk by using $|\cdot |$ or $max\\{0,\cdot\\}$.
>
> Lu, N., Zhang, T., Niu, G., and Sugiyama, M. Mitigating overfitting in supervised classification from two unlabeled datasets: A consistent risk correction approach. In International Conference on Artificial Intelligence and Statistics, pp. 1115–1125. PMLR, 2020.
>
> &nbsp;
>
> **Q4. Would consistency regularization term lead to more instance pairs with smaller confidence differences, thereby increasing the presence of noisy supervised signals?**
>
> Thank you for your comments. The core objective of consistency regularization is to enforce consistency in the classifier's predictions for pairwise instances with small confidence differences by constraining the model output. It is important to clarify that our optimization target is the classifier's output $ g(\cdot)$, not the confidence difference $c$. In our setting, $c$ serves as an attribute used for training, functioning as a form of weak supervision. Our goal is not to optimize $c$, but to use it to guide the classifier toward the desired outputs. Therefore, the concern about "causing more pairs to have smaller confidence differences" does not apply here.

---

### Official Review · Reviewer_UWhK · 2025-03-17

**Overall Recommendation:** 3

**Summary:**

The paper studies a special type of weakly-supervised learning known as confidence difference learning. This method leverages confidence differences between unlabeled data pairs to improve classifier training under noisy real-world conditions. By incorporating a noise generation technique and a risk estimation framework that includes consistency risk and regularization, ConfDiff classification demonstrates enhanced robustness and outperforms traditional methods in experiments on benchmark and UCI datasets. Theoretical analyses providing error bounds for the risk estimations further support the method's effectiveness.

**Claims And Evidence:**

yes

**Essential References Not Discussed:**

n.a.

**Experimental Designs Or Analyses:**

Yes.

**Methods And Evaluation Criteria:**

yes

**Other Comments Or Suggestions:**

n.a.

**Other Strengths And Weaknesses:**

1. The main theoretical contribution of this paper improved over [1] seems to be the incorporation of consistency regularization and its effect in error bound; based on my understanding, $C_{g}$ bounded the differences between $x$ and $x'$, so that under authors' setup, if the prediction of these data points is close enough, then the perceived generalization error should decrease, which is sensible.

2. After a very coarse examination, the proof of this paper seems to be correct.

3. Encourage instances with smaller confidence differences to produce similar outputs that seem intuitive and sensible, both theoretically and empirically.

[1] Binary classification with confidence difference, NeurIPS 2023.

Weaknesses:

1. I feel the motivation of this paper is not strong enough, and I cannot see much real-world scenerios that motivates this problem.

2. It seems that this problem is only applicable for binary classification, which further limits its application in real-world scenerios.

**Questions For Authors:**

n.a.

**Relation To Broader Scientific Literature:**

n.a.

**Theoretical Claims:**

Claims seems to be correct.

---

> ### Author Rebuttal · Authors · 2025-03-31
>
> **Q1. The motivation of this paper is not strong enough, and what real-world scenerios can motivates this problem?**
> Thank you for your suggestions.
>
> **About motivation.** The motivation for our method arises from the observation that small confidence differences may lead to imprecise guidance within $R_{CD}$, particularly when the confidence difference equals zero, resulting in a complete lack of predictive guidance. Our experiments in Figure 1 further demonstrate that pairwise instances with larger confidence differences dominate the contribution to $R_{CD}$, while those with smaller confidence differences contribute minimally. To address this issue, we propose a consistency regularization term that encourages $\mathbf{x}$ and $\mathbf{x'}$ to produce more similar outputs in the model when $|c(\mathbf{x}, \mathbf{x'})|$ is small.
>
> **About real-world scenerios.**
>
> 1. Rehabilitation assessment. The assessment of whether a patient meets rehabilitation criteria presents significant challenges. Individual differences in recovery make the evaluation process inherently subjective. In addition, assessments become particularly uncertain when a patient is close to the threshold of rehabilitation. In contrast, clinicians can more reliably generate approximate confidence labels using objective data, such as scores from functional assessment scales (e.g., FIM or Barthel Index), questionnaire responses, and motor performance metrics. Moreover, by considering the continuity of the rehabilitation process and individual differences, changes in a patient's condition over time can be used to construct confidence difference that reflect recovery trends, leading to more robust rehabilitation assessment.
>
> 2. Click-through rate prediction. In recommender systems, predicting whether a user will click on a given item is a central task. However, due to the sparsity of click data and the problem of class imbalance, it is often difficult to assign accurate pointwise label to each user-item pair. In contrast, collecting pairwise preference information, which refers to the relative preference between candidate items for a given user, is a more feasible and effective alternative. In practical applications, approximate confidence can be obtained using auxiliary probabilistic classifiers, such as models predicting click probabilities. Moreover, in many recommendation scenarios such as news, short video, and movie recommendations, real-valued feedback like watch ratio or user ratings is often available, which can be leveraged to construct more informative confidence difference.
>
> In addition, it also holds practical value in various domains such as obstacle detection in autonomous driving, driving behavior analysis, and financial risk assessment.
>
> &nbsp;
>
> **Q2. It seems that this problem is only applicable for binary classification, which further limits its application in real-world scenerios.**
>
> Thank you for your insightful comments. The method proposed in this paper is indeed developed within the framework of binary classification. Nevertheless, we respectfully believe that this setting does not substantially limit its applicability to real-world scenarios.
> First, a wide range of real-world applications naturally take the form of binary classification problems, where our method can be directly implemented. Typical examples include medical diagnosis, rehabilitation assessment, and financial risk management.
> Furthermore, the proposed method is conceptually general and can be naturally extended to multi-class classification tasks, which we consider a promising direction for future research. Formally，let $ c_i \in \mathbb{R}^l $ be the confidence difference between pairwise unlabeled data $(\mathbf{x}_i,\mathbf{x}'_i)$ drawn from an independent identically distribution probability density $p(\mathbf{x},\mathbf{x}')=p(\mathbf{x})p(\mathbf{x}')$:
>
> $$ \mathbf{c}_i = [c_i^{(1)}, c_i^{(2)},\dots ,c_i^{(l)}],\\:\\:c_i^{(k)}=c^{(k)}(\mathbf{x}_i,\mathbf{x}'_i)=p(y'_i=k|\mathbf{x}'_i)-p(y_i=k|\mathbf{x}_i) $$
>
> where $l$ denotes the number of classes. Accordingly, the consistency regularization term over $D^{C}$ in the expected risk can be modified as:
>
> $$\alpha \mathbb{E}\_{p\_{{\small \mathcal{D}^{C}} }(\mathbf{x},\mathbf{x}')} [ \bigl(\frac{1}{ \log \left(\left| \mathbf{c(\mathbf{x},\mathbf{x}')}\right|_1 + \varepsilon  \right)  } \bigr)  \cdot  \left \\| g(\mathbf{x})-g(\mathbf{x}') \right \\|_2 ] $$
>
> In summary, this work proposes a general framework for addressing noisy supervision signals in confidence difference classification. The proposed method is not limited to binary classification tasks and also demonstrates practical applicability in real-world scenarios.

---

> > ### Comment · Reviewer_UWhK · 2025-04-06
> >
> > Thanks for authors' detailed comments, I will maintain my evaluation as leaning towards acceptance.

---

### Decision · Program_Chairs · 2025-05-01

**Decision:**

Accept (poster)

**Comment:**

Three reviewers support acceptance, highlighting the work's novelty, theoretical rigor, and empirical validation. However, Reviewer joRe, who evaluated this paper at ICLR 2025, notes that the current submission does not incorporate updates beyond the ICLR'25 final version (which already addressed their concerns during ICLR'25 rebuttal). So, I recommended Reject in the previous stage.

After discussion with the SAC, I have compared the current submission with its ICLR'25 versions (both initial and revised version after ICLR'25 rebuttal). The authors did address most of the concerns during the ICLR'25 rebuttal, as the current manuscript differs significantly from the ICLR'25 initial version, with substantive revisions made during the rebuttal process. Therefore, I now lean toward Weak Accept, as the core criticisms were previously mitigated.